# ACTION AS A MODALITY: TURNING MULTI-MODAL LLMS TO GENERAL ACTION PLANNERS

## ABSTRACT

Large Language Models (LLMs) have demonstrated strong reasoning capabilities and possess extensive common knowledge. This enables them to adapt to a variety of complex tasks in a zero-shot manner, including functioning as controllers to manipulate automated systems and produce executable action sequences. However, a significant challenge in the existing framework is the misalignment between the general pre-trained LLM and the action space of specific control tasks. This misalignment necessitates extensive efforts in designing task-specific prompts, which are less generalizable and do not ensure consistent output when prompting a pre-trained LLM to generate the desired action sequences. To address this issue, we propose a novel solution, ActionVerse, which encodes action candidates into a series of modality tokens, coupled with an efficient alignment technique to synchronize the action tokens with the LLM's language space. By leveraging this approach, the proposed ActionVerse successfully transforms a chat-based multi-modal LLM into a general action executor capable of handling tasks requiring step-by-step execution of various actions. Experiments on several sequential action tasks demonstrate the effectiveness of the proposed framework.

## 1 INTRODUCTION

Large Language Models (LLMs) (Radford et al., 2018; Touvron et al., 2023a;b; Radford et al., 2019; Brown et al., 2020) trained on vast amounts of textual data have demonstrated strong reasoning capabilities for completing complex tasks based on human instructions. These emerging capabilities can be effectively generalized to a variety of text-based tasks in a zero-shot or few-shot manner, significantly promoting the application of LLMs in various domains, such as chatbots (OpenAI, 2023), code copilots (Li et al., 2022), and more.

Recently, benefiting from Parameter-Efficient Fine-Tuning (PEFT) techniques (Hu et al., 2022; Dettmers et al., 2023), it has become computationally affordable to adapt text-based LLMs to a multi-modal scenario by tuning the foundational LLM with a few aligning projection layers as well as rank decomposition matrices. This unlocks the potential for text-based LLMs to perceive and generate non-textual content with minimal tuning (Liu et al., 2023; Ye et al., 2024; Zhu et al., 2023; Wu et al., 2023). The ModaVerse (Wang et al., 2024) is one of the representative examples. Equipped with the capabilities of reading, seeing, and hearing, these Multi-modal Large Language Models (MLLMs) can be applied to broader scenarios like controlling smart devices that pure text-based models cannot handle, such as navigating a robot based on camera inputs (Zhou et al., 2024; Zhang et al., 2024b), automatically controlling smartphone apps (Zhang et al., 2023a; You et al., 2024), and manipulating robots (Zhang et al., 2023b; Yang et al., 2023b; 2024). These methods leverage the extensive common knowledge embedded within MLLMs to analyze multi-modal inputs, such as a robot's camera feed, GPS location signals, and mobile phone screenshots, to comprehend the environment. Based on this understanding, they generate a set of well-reasoned operational steps that enable the devices to execute actions and achieve the desired objectives. For instance, NavGPT (Zhou et al., 2024) proposes a zero-shot pipeline that employs ChatGPT for explicit reasoning and generating navigation routes for robots, while AppAgent (Zhang et al., 2023a) introduces a framework that relies on GPT-4v to operate smartphone applications.

However, the fact that the foundational MLLMs were trained for general purposes results in a redundant tokenizer, creating an inevitable gap between the text space and the desired action space of

the controlling tasks. Therefore, current frameworks typically design complicated system prompts with detailed instructions and sample outputs to guide the MLLMs in generating the expected format of outputs. Despite these efforts, several issues remain with significant room for improvement: (1) The nature of the tokenization process in MLLMs means that the expected output format may not always be achieved, especially when complex non-text action indices are involved. For example, producing an action command that asks the robot to move to a viewpoint indexed by 'b0a6cf6e9b904324...' can be problematic sometimes. (2) The extremely long system prompts are difficult to design and maintain, often requiring expert manual effort, and they significantly increase the number of input tokens during the inference stage, thereby considerably raising computation costs. (3) Such designs make it difficult to utilize the history of action sequences to refine decision-making and introduce a self-correction mechanism.

Recently, some works (Chen et al., 2023a; Bai et al., 2023; You et al., 2023; Ma et al., 2024) have explored incorporating object-level visual tokens with MLLMs to enhance visual grounding and referring capabilities. Specifically, these methods typically follow a spatial-wise sampling process to extract candidate object areas in the input images. The image features of each candidate region are encoded as visual tokens, which are then inserted into the input tokenized sequence for MLLMs, enabling MLLMs to refer to certain areas by providing the coordinates on the images. Inspired by this, to address the aforementioned issues in current MLLMs in generating action-oriented outputs, we designed an action encoder to explicitly tokenize the action candidates into a series of action tokens, which are then extended into the input sequence of the MLLMs. To align this action feature space with the foundational LLM's language space, we conduct instruction tuning with a few projection layers and PEFT techniques, thus enabling efficient adaption to turn chat-based MLLMs into action executors. Within this design, the proposed framework is no longer limited to using textual descriptions to represent the action indices but uses a series of unified action tokens such as <ACTION1>, <ACTION2> to represent the action candidate with rich features. This approach is easily adaptable to more generalized scenarios requiring MLLMs to produce action operation orders, avoiding the misalignment between the text space of MLLMs and the action space of controlling tasks. It also eliminates the need for overly complicated system prompts and benefits from the natural ability of chat-based LLMs to use context information to refine the decision-making process. Therefore, to train the proposed method, we introduce a pipeline to build a conversation-like instruction-following training set, which also incorporates a data augmentation method that introduces disturbances into the routes to facilitate an implicit self-correction capability. In summary, this paper presents an efficient paradigm to adapt chat-based MLLMs to generate executable action sequences for controlling smart agents. Specifically, the contribution of this work is threefold:

- We propose *ActionVerse*, a general framework designed to adapt MLLMs to produce executable action sequences for automatic systems. ActionVerse consists of two core components: an action encoder that tokenizes the action candidates into a series of action tokens, and a projection module that translates these tokens into the language space interpretable by the foundational LLM. With proper instructions, ActionVerse can select and produce the appropriate action sequence for specific controlling tasks.

- Controlling tasks typically expect a sequence containing multi-step actions to guide the smart agents to fulfill specific goals, matching the nature of multi-round conversations in chat-based LLMs. Therefore, we proposed a conversation-like instruction-following tuning procedure for training the ActionVerse, which incorporates an implicit self-correction mechanism to rectify possible wrong movements executed in previous steps.

- To evaluate the effectiveness of ActionVerse, we conduct experiments on two controlling tasks: robot navigation (Anderson et al., 2018) and website navigation (Chen et al., 2024b). Experimental results demonstrate that within minimal fine-tuning, ActionVerse achieves performance comparable to state-of-the-art methods.

## 2  RELATED WORK

**Action-oriented LLMs.** The strong reasoning abilities of LLMs enable them to perform complex tasks based on human instructions, making them ideal for embodied tasks (Zeng et al., 2023) that require reasoning and common knowledge (Zhang et al., 2023a; Zhou et al., 2024; Lu et al., 2023; Mu et al., 2023; Yang et al., 2023b; Zhang et al., 2023b). In these tasks, LLMs act as agents,

understanding tasks and environments to plan actions towards a goal, enabling applications like robotic manipulation and smartphone control. For example, RT-2 (Brohan et al., 2023) extends LLMs to robotic control but requires extensive trajectory data, while LaMo (Shi et al., 2023) handles simpler scenarios like Atari and MuJoCo using time-series data. AppAgent (Zhang et al., 2023a) leverages ChatGPT-4v for smartphone app control, and NavGPT (Zhou et al., 2024) uses ChatGPT to plan robot navigation routes. LLaRP (Szot et al., 2023) applies LLMs to online reinforcement learning, and RoboFlamingo (Li et al., 2023) frames video observations in a POMDP setting.

Most methods, however, depend on commercial LLM APIs (OpenAI, 2023), like ChatGPT or GPT-4v/o, to produce action-oriented text. While avoiding LLM training, this introduces challenges, such as complex system prompts. For instance, NavGPT (Zhou et al., 2024) and AppAgent (Zhang et al., 2023a) use prompts up to three thousand words, which are hard to maintain as APIs evolve and incur high costs due to expensive API calls. Additionally, requiring internet access limits real-time applications in robotics. To mitigate these issues, some works (Lu et al., 2023) train or fine-tune LLMs for action outputs, like Unified-IO 2 (Lu et al., 2023), which manipulates robotic arms using over a million trajectories. However, these methods demand substantial resources, highlighting the need for more efficient solutions that balance LLM-based agents and training from scratch.

**Multi-modal LLMs.** MLLMs (Zhang et al., 2024a) extend text-based LLMs to perceive and generate non-text content. Mainstream MLLMs can be categorized into two types: native multi-modal LLMs (Team, 2024; Lu et al., 2023; Reid et al., 2024; McKinzie et al., 2024) and adapted multi-modal LLMs (Liu et al., 2023; Zhu et al., 2023; Wang et al., 2024; Ye et al., 2024). Native MLLMs are inherently designed to handle diverse data modalities, such as original image pixels and audio signals. This design preserves high-fidelity details of multi-modal inputs, allowing for more natural and efficient multi-modal understanding and generation. However, it requires extensive training resources, including massive paired multi-modal data and significant computational power. For instance, Unified-IO 2 (Lu et al., 2023) pre-trains an auto-regressive MLLM that can interpret and generate multiple modalities involving vision, language, audio, and action using an extremely large training set containing over 1 trillion text tokens, 1 billion image-text pairs, and millions of videos, 3D assets, and agent trajectories. This pre-trained foundation model is then further fine-tuned using an instruction-following set that combines more than 120 datasets covering 220 tasks across vision, language, audio, and action. Similarly, Chameleon (Team, 2024) also pre-trains on an extensive training set containing nearly 3 trillion text-only tokens, 1.5 trillion text-image tokens, and 400 billion tokens of interleaved text-image data. On the other hand, instead of training an MLLM from scratch with a mixture of multi-modal data, adapted MLLMs aim at adapting a text-based foundational LLM into a multi-modal scenario. For example, Flamingo (Alayrac et al., 2022) follows a late-fusion design, where the multi-modal inputs are encoded by a series of separate encoders before being combined together. Similarly, MiniGPT-4 (Zhu et al., 2023) employs a Q-former to encode images and then translates the visual features to the LLM space through a linear projection layer. NextGPT (Wu et al., 2023) and ModaVerse (Wang et al., 2024) further extend this paradigm to the output space, enabling any-to-any modality transformation. Such designs enable text-based LLMs to interpret and generate non-text content with minimal fine-tuning on limited datasets, offering a more efficient solution for MLLMs. More recently, fine-grained comprehension and open-world referring and grounding have become important for evaluating the reasoning capabilities of visual-focused MLLMs. These methods (Peng et al., 2023; Chen et al., 2023b; Ma et al., 2024; You et al., 2023; Bai et al., 2023) are designed to comprehend and answer questions related to specific regions of interest in images using user input bounding boxes or hybrid region representations, and refer to these areas by outputting the coordinates of specific regions. For example, Ferret (You et al., 2023) accommodates a broad range of free-form shapes for referring, including points, bounding boxes, sketches, scribbles, polygons, and more, facilitating more fine-grained question-and-answer interactions with MLLMs. However, despite extensive efforts in MLLM research, most work has focused on regular modalities such as image, video, and audio, while a more efficient way to adapt these MLLMs to the action space remains underexplored.

**Efficient LLMs.** LLMs, typically comprising tens to hundreds of billions of parameters, require vast amounts of data and computational resources when trained from scratch. PEFT techniques (Hu et al., 2022; Dettmers et al., 2023) have emerged as a more cost-effective alternative for fine-tuning these models. Instead of fully fine-tuning the models, PEFT techniques generally freeze most of the parameters and tune only a small fraction, typically less than 1% of the entire model, thus significantly reducing training costs. To address the shortage of training data, one practical approach

is to use commercial LLMs like ChatGPT (OpenAI, 2023) to generate high-quality instruction-following data (Zhu et al., 2023; Liu et al., 2023), facilitating fine-tuning for downstream tasks. Recently, combining these training efficiency and data efficiency techniques has become a widely-used strategy for developing new LLM-based methods, allowing for more affordable adaptation of LLMs to a broader range of tasks and scenarios. Additionally, during the deployment phase, various post-training compression techniques (Frantar et al., 2023; Dettmers et al., 2022; Lin et al., 2024) have been explored to enhance inference speed and optimize performance on resource-limited devices. Given the sparse nature of LLMs, these techniques typically involve compressing the models through pruning or quantization using a calibration dataset. This allows LLMs to be deployed on a significantly smaller scale while maintaining performance comparable to full-precision models, thus greatly reducing computational overhead.

## 3 METHOD

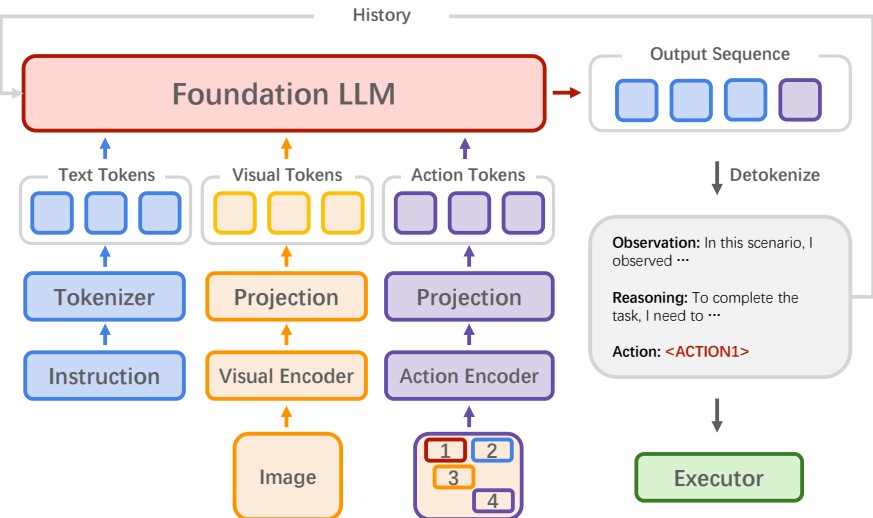

Figure 1: Pipeline of ActionVerse. The three streams respectively encode the text, visual, and action tokens, which are then merged together by the foundation LLM to produce the output sequence. This sequence includes both the reasoning process and the proposed action for the current time step. The actions are then parsed and sent to the executor for execution. This process operates in a conversation-like manner, maintaining a context history to refine future decision-making.

**Action planning tasks.** Unlike generic MLLMs that generate general textual responses in natural language, the action planning task requires the planner $\pi(\cdot)$ to produce a sequence $S$ consisting of operations $\{s_1, s_2, s_3, \ldots, s_t\}$, where $s_t \in \mathcal{A}$ and $\mathcal{A}$ contains actions executable by intelligent agents, *e.g.*, MoveForward() or TurnLeft(). To produce such a sequence, the planner typically has access to observations $\mathcal{O} = \{o_1, o_2, o_3, \ldots, o_t\}$ from the environment. At each time step $t$, the observation $o_t$ may consist of various signals, such as images from different angles, GPS signals, depth information, *etc.*, represented as $< o_t^1, o_t^2, o_t^3, \ldots, o_t^n >$. Based on the observations $\mathcal{O}$ and specified instruction $\mathcal{I}$, the planner is able to make a single-step inference to decide which action should be taken at the next time step, denoted as $\pi(s_{t+1} \mid o_t, \mathcal{I})$. Furthermore, by incorporating the history of action trajectories $s_{1:t}$, it is also possible for such planners to refine their predictions given the operation histories, denoted as $\pi(s_{t+1} \mid o_t, s_{1:t}, \mathcal{I})$. For MLLM-as-agent-based methods, the generic MLLM serves as the planner $\pi(s_{t+1} \mid o_t, s_{1:t}, \mathcal{I}; \Theta)$, where $o_t$ is typically the camera feed or screenshots that show the current environment, $s_{1:t}$ is the chat history, and $\mathcal{I}$ is the instruction accompanied by the system prompts. The weights $\Theta$ are trained on a general corpus without specific fine-tuning on downstream data.

### 3.1 ACTIONVERSE

ActionVerse is a generic framework designed for action planning tasks, which maps the action space of specific control tasks to the foundation MLLMs using an action encoder and a series of projection layers. As shown in Figure 1, ActionVerse consists of three main components: first, the text instruction is tokenized into a series of text tokens; second, a visual encoder encodes the visual observations and then maps them into visual tokens via projection layers; and finally, an action encoder encodes the action candidates at the current time steps and maps them into action tokens through projection layers. The aim of this work is to adapt MLLMs to the action space efficiently. We built ActionVerse upon existing chat-based MLLM frameworks (Liu et al., 2024), following the adaptation training of the visual encoder as well as the visual projection layers. The main difference lies in the newly introduced action encoders that align the action space with the foundational LLM, which will be introduced in Section 3.1.1. Additionally, to explicitly direct the model to perform a Chain-of-Thought-like (CoT-like) reasoning process, ActionVerse is capable of outputting observations based on the given images as well as the reasoning process to fulfill the given instructions. By incorporating historical outputs, it has learned an implicit capability to correct incorrect previous movements. We introduce the details of building the multi-round conversation-like training set in Section 3.1.2.

### 3.1.1 ACTION ENCODING

**Action space.** In control scenarios, the action set $\mathcal{A}$ consists of operations that can be executed by specific devices, which may vary from task to task. Typically, however, the action types for a particular device are discrete and limited in number, while each of these actions may further take discrete or continuous arguments as inputs. For example, for an embodied robot, the action set may include operations like `Move()`, `Pick()`, and `Turn()`, whereas a virtual assistant on a smartphone may have actions like `Text()`, `Click()`, and `Drag()`. Additionally, each specific action, such as `Turn()`, may further take arguments for fine-grained operations, either discrete directions such as *left* and *right*, or a continuous numeric value representing the degree of rotation.

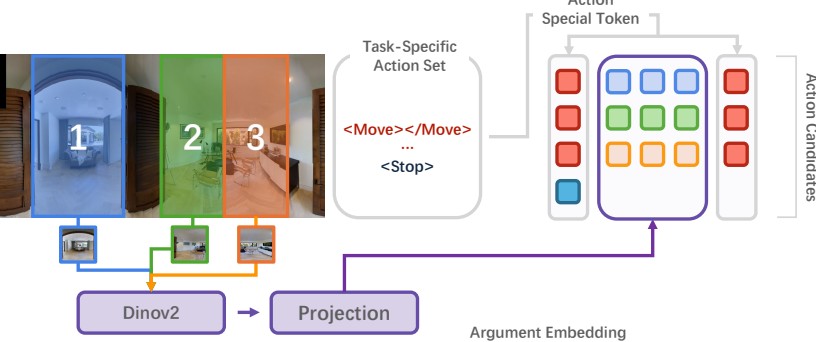

Figure 2: Action encoding process in ActionVerse. Using a vision-language navigation (Anderson et al., 2018) task as an example, DINOv2 is used as the action encoder in this case to extract the visual features of each navigable region as the argument embedding. These embeddings are then inserted between task-specific special action tokens, such as <Move> and </Move>, forming a candidate pool that offers all executable operations at the current timeframe, such as `Move(ViewPoint1)`, `Stop()`, *etc*.

**Existing action encoding.** In existing MLLM-based control methods (Zhou et al., 2024; Chen et al., 2024a), a typical solution for encoding these actions and arguments relies solely on *text descriptions*. The actions and arguments are usually defined with examples in the system prompts, then tokenized and treated like plain English words. However, such frameworks may lead to several issues. (1) It necessitates extensive work to design and maintain the system prompts; (2) Due to the context window length capacity, an overly long system prompt will reduce the number of tokens available for other inputs; (3) Without adding special tokens for actions in the vocabulary, it is necessary to maintain the system prompts of definitions of the actions in multi-round interactions, which

wastes considerable resources on processing repeated lengthy prompts; (4) Due to tokenization, text descriptions of actions or arguments tend to be separated into pieces. For example, in some previous works (Zhou et al., 2024) that directly ask ChatGPT to encode or generate long action IDs such as '68aafa779b9c41eca16156cfddcedd2b', the result may be a token sequence like ['6', '8', 'a', 'af', ....]', which can break the original semantics and result in potential distortion during the next token generation process in MLLMs, and increases the computational burden of processing more tokens. Another approach that partially alleviates the above issues involves introducing *visual prompts* within the inputs. This approach leverages the emerging grounding capability of MLLMs when explicit marks are given (Yang et al., 2023a). It involves adding specific visual labels to indicate regions that may contain executable elements (Zhang et al., 2023a), thus avoiding the need for complex text descriptions of actions. This simple yet effective strategy bridges the gap between actions and visual observations. However, it presents challenges for the visual recognition abilities of MLLMs, such as OCR, to accurately identify the provided marks, especially when they are densely packed. Furthermore, although this approach seems promising for actions that do not require complicated arguments, like Click() buttons on smartphones, it remains difficult to incorporate arguments by embedding visual marks into the images, thus limiting its applicability in more generalized scenarios.

**ActionVerse.** To address the above challenges in action encoding for MLLMs, it is necessary to solve two problems: first, to design a more efficient way to represent the action space, and second, to embed the argument for each action. While the most straightforward approach is to design new action tokenizers and train an action-based foundational MLLM (Lu et al., 2023), this consumes too many training resources. Inspired by recent success (Wang et al., 2024) in efficiently adapting text-based LLMs into multi-modal scenarios by tuning only a few projection layers to align non-textual features with the text space of LLMs, we have designed a similar pipeline. Specifically, as shown in Figure 2, instead of using text descriptions or tagging the executable regions on images with numeric marks, we encode the features of operable areas and incorporate task-specific special action tokens to form a candidate pool of the current executable actions. By doing this, it simultaneously encodes the action and corresponding arguments, which are further mapped into the original MLLM space by some linear projection layers. Therefore, it becomes capable of embedding action candidates into the input sequence, serving as a guidebook for producing expected action sequences without specifying lengthy system prompts.

Taking a typical action-oriented task Vision Language Navigation (VLN) (Anderson et al., 2018) as an example, we formally introduce the action encoding process. In VLN, an embodied robot is asked to fulfill an instruction $\mathcal{I}$ described in natural language in a simulated environment. The robot itself can perceive the environment through equipped sensors, such as an RGB camera and a depth camera. Based on these observations, the robot is able to make two actions, Move() and Stop(), which constitute the action space $\mathcal{A} = \{\text{Move()}, \text{Stop()}\}$ of this task. Specifically, the Move() action further takes a ViewpointID as input to teleport to a specific location in the simulator, which is a simplified version of movement compared to the real world. The other action, Stop(), should be taken when the agent decides it has navigated to the target position to finish the task, which does not need any input arguments. We first build the task-specific action set $\mathcal{D}$, which is a set of special tokens that need to be added to the vocabulary of the foundational MLLM tokenizer. Each special token matches a specific action in $\mathcal{A}$. There are two types of special tokens in $\mathcal{D}$: the first represents the actions that take arguments as input, *e.g.*, Move(), which is represented by paired tokens like <Move> and </Move>, while others represent actions that do not need inputs, such as Stop(), which is represented by a single token like <Stop>. To encapsulate the Move() action's argument, which includes direction information in the VLN task, we use DINOv2 (Oquab et al., 2024) to extract visual features of navigable regions (see Figure 2). Each navigable viewpoint in the panorama image of the current location is cropped into patches and sent into vision transformers to extract vision features. These features are then translated into argument embeddings through linear projection layers, inserted between the <Move> and </Move> tokens. Since the Stop() action does not require any arguments in this task, there is no argument embedding for this special token.

Although we use the VLN task as an example here to explain the action encoding framework in ActionVerse due to its simplicity, it is important to note that the proposed framework can be seamlessly extended to more generalized scenarios. For example, the visual observations can be extended to broader modalities such as depth and thermal graphs; the viewpoint regions can be transferred to buttons on smartphones, robotic arms, etc.

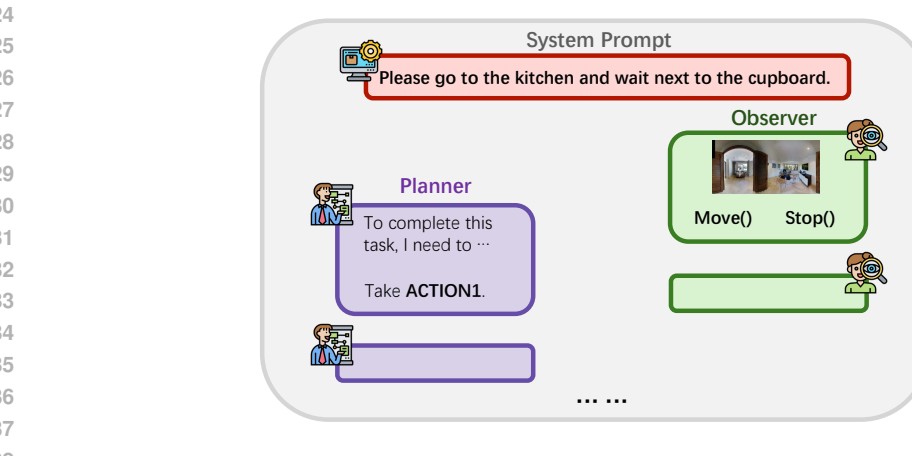

Figure 3: Conversation-like interaction between the observer and the planner: at each time step, the observer inputs observations along with the candidate actions, while the planner responds with a reasoning process based on the observations and system prompts, proposing the actions to be taken.

### 3.1.2 CONVERSATION-LIKE ACTION INSTRUCTION-FOLLOWING TUNING

In control scenarios, the planner is typically expected to produce a sequence of actions in continuous time steps until the goal is achieved. At each time step, the planner receives new observations as inputs and is expected to propose the movement for the next time step. This is analogous to a conversation between the observer and the planners, mirroring the nature of chat-based MLLMs. Therefore, to efficiently train ActionVerse, particularly the projection layers and adaptors of the foundational MLLMs, and to align with the newly introduced action space, we propose a pipeline to build a conversation-like instruction dataset for action-based MLLMs, following recent practices in the efficient training of MLLMs (Liu et al., 2023; Wang et al., 2024; Zhu et al., 2023).

Specifically, as shown in Figure 3, we break down the operation flow into a multi-round conversation. At the beginning of each conversation, the instruction is set as the system prompt, which states the target objective to be achieved by the agent. Then, a multi-round conversation follows between the observer and planner. At each time step, the observer inputs the current observations as well as the action candidates, while the planner responds to the observer with a reasoning process and proposes the action to be taken. The conversation will continue until the planner terminates the interaction upon achieving the goal. Following this pipeline, we build task-specific conversation-like action instruction training sets, which are used to fine-tune the MLLMs and projection layers. To generate the detailed reasoning processes of the planners and textual descriptions of observations in the conversations, we employ the ChatGPT APIs, as detailed in the Appendix A.

Moreover, to enhance the robustness of action sequence generation, we introduce disturbances in the routes within the conversation datasets as a data augmentation strategy. For example, for a ground-truth action sequence A->B->C, we might add an incorrect step from B to D, then self-correct back to B and proceed to the target C: A->B->D->B->C. This method introduces an implicit self-reflection capability that rectifies erroneous movements.

## 4 EXPERIMENTS

### 4.1 TASK SETTING

To evaluate the effectiveness of the proposed ActionVerse, we conducted experiments on two action-based tasks: Room-to-Room Vision Language Navigation (R2R-VLN) (Anderson et al., 2018) and Web Vision Language Navigation (Web-VLN) (Chen et al., 2024b). Implementation details including the model structure, training hyperparameters, and training procedures, are provided in Appendix B.

**R2R-VLN** is a classic action-based task where an embodied robot in a simulator is instructed to navigate from the initial point to a specific location. At each position, the robot can observe the

environment through both an RGB camera and a depth camera. The action space of this task is greatly simplified: the robot is allowed to teleport to any reachable viewpoint within a certain distance without needing to consider obstacles or specific routes. When the robot observes that it has reached the target location, it is expected to execute a stop action.

**Web-VLN** is a web-based version of the VLN task, simulating web browsing activities. Specifically, given a language instruction, the agent's goal is to navigate across different websites to find specific elements. For example, it may need to search for a certain commodity on a shopping website. Web-VLN shares a similar setting and evaluation metrics with R2R-VLN, though the action space varies from moving to specific viewpoints among rooms to clicking specific buttons on websites.

**Metrics.** R2R-VLN and Web-VLN utilize similar metrics, including Trajectory Length (TL), which measures the average distance the agent travels; Navigation Error (NE), indicating the average distance from the agent's final position to the target; Success Rate (SR), representing the percentage of navigation attempts where the agent successfully reaches the target location within a 3-meter margin of error; Oracle Success Rate (OSR), which tracks the agent's success rate when stopping at the closest point to the goal along its path; and Success rate weighted by Path Length (SPL), which evaluates the balance between navigation accuracy and efficiency by adjusting the success rate based on the ratio of the optimal path length to the agent's actual path length.

## 4.2 QUANTITATIVE RESULTS

We compare our method with state-of-the-art techniques on both the R2R-VLN and Web-VLN datasets. This includes both VLN expert models (Anderson et al., 2018; Fried et al., 2018; Tan et al., 2019; Hao et al., 2020; Hong et al., 2021; Chen et al., 2021; 2022) specifically designed for the tasks and MLLM-as-agent methods (Zhou et al., 2024; Tan et al., 2019).

Table 1: Comparison with state-of-the-art methods on R2R-VLN task.

| Method Category | Method | TL↓ | NE↓ | OSR↑ | SR↑ | SPL↑ |
|---|---|---|---|---|---|---|
| Expert Model | Seq2Seq (Anderson et al., 2018) | **8.39** | 7.81 | 28 | 21 | - |
| | Speaker Follower (Fried et al., 2018) | - | 6.62 | 45 | 35 | - |
| | EnvDrop (Tan et al., 2019) | 10.70 | 5.22 | - | 52 | 48 |
| | PREVALENT (Hao et al., 2020) | 10.19 | 4.71 | - | 58 | 53 |
| | VLN-BERT (Hong et al., 2021) | 12.01 | 3.93 | 69 | 63 | 59 |
| | HAMT (Chen et al., 2021) | 11.46 | **2.29** | 73 | 66 | **61** |
| | DuET (Chen et al., 2022) | 13.94 | 3.31 | **81** | **72** | 60 |
| LLM-as-agent | DuET (Init. LXMERT (Tan & Bansal, 2019)) | 22.03 | 9.74 | 7 | 1 | 0 |
| | NavGPT (Zhou et al., 2024) | **11.45** | 6.46 | 42 | 34 | 29 |
| | MapGPT (GPT-4) (Chen et al., 2024a) | - | 6.29 | 58 | 39 | 26 |
| | MapGPT (GPT-4v) (Chen et al., 2024a) | - | 5.63 | 58 | 44 | 35 |
| | ActionVerse (ours) | 11.93 | **4.12** | **66** | **60** | **55** |

Table 1 presents the quantitative results for the R2R-VLN task. Our method, ActionVerse, demonstrates competitive performance compared to other LLM-as-agent methods, achieving superior results in NE, OSR, SR, and SPL metrics. However, there remains a performance gap between our approach and the expert models specifically tailored for this task. This gap highlights that significant improvements are needed for LLM-as-agent methods to reach the proficiency of specialized expert models, indicating potential issues and future research directions in this domain.

The expert models outperform LLM-based methods largely due to their meticulous design and reliance on fine-grained information such as GPS signals, object-level annotations, and topological mappings of rooms. These models are finely tuned to handle the specific challenges of VLN tasks, enabling them to achieve higher accuracy. In contrast, ActionVerse follows a generic design for action-based tasks without additional enhancements, relying solely on general visual observations and the reasoning capabilities of MLLMs.

When compared with other LLM-as-agent methods, ActionVerse achieves better performance. This improvement can be attributed to our alignment pipeline, which simultaneously encodes system prompts, visual observation tokens, and action tokens within the input sequence. This design allows for more fluent and coherent prediction of action sequences. Other agent-based methods, such as NavGPT and MapGPT, convert visual observations into text descriptions before feeding them into

the LLM along with system prompts. This process inevitably loses rich visual information, resulting in relatively lower performance.

Table 2: Comparison with state-of-the-art methods on Web-VLN task.

| Method Category | Method | TL↓ | OSR↑ | SR↑ | SPL↑ |
|---|---|---|---|---|---|
| Expert Model | VLN-BERT (Hong et al., 2021) | 6.96 | 18.62 | 18.62 | 18.14 |
| | WebGUM (Chen et al., 2021) | **3.44** | 31.78 | 31.22 | 31.22 |
| | WebVLN-Net (Chen et al., 2024b) | 3.71 | 39.54 | 39.46 | 39.46 |
| LLM-as-agent | NavGPT (Zhou et al., 2024) (GPT-3.5-turbo) | 4.98 | 12.94 | 7.46 | 4.53 |
| | NavGPT (Zhou et al., 2024) (GPT4) | 5.43 | 21.89 | 16.92 | 11.61 |
| | ActionVerse (ours) | 5.62 | **41.18** | **40.02** | **39.78** |

Table 2 shows the results on the Web-VLN task. As this is a relatively new benchmark, existing methods are mostly adaptations of R2R-VLN-based models. Our method achieves the best performance among the compared methods, indicating its effectiveness in this domain. The superior performance of ActionVerse suggests that it can better handle the visual and navigational complexities of web environments.

It is noteworthy that NavGPT, which utilizes ChatGPT APIs, obtains a relatively low score on this task. This could be due to its approach of transferring images into text descriptions, which may not fully capture the nuances of website layouts. This observation underscores the importance of preserving rich visual information in VLN tasks and suggests that methods relying solely on text descriptions may face limitations.

In summary, our proposed ActionVerse shows that with proper alignment and integration of visual and action information, MLLMs can achieve competitive performance on VLN tasks. While there is still room for improvement to match the specialized expert models, our findings encourage further exploration into refining LLM-based methods to bridge this gap. Future work could focus on incorporating domain-specific knowledge and fine-tuning strategies to enhance the capabilities of LLM-as-agent methods in vision-and-language navigation tasks.

### 4.3 ABLATION STUDY

Table 3: Ablation study on R2R-VLN task.

| **Without Conversation-like Data** | | | | | |
|---|---|---|---|---|---|
| Method | TL↓ | NE↓ | OSR↑ | SR↑ | SPL↑ |
| Plain | 11.03 | 4.71 | 57 | 52 | 50 |
| +Depth | 11.05 | 4.62 | 58 | 54 | 51 |
| **With Conversation-like Data** | | | | | |
| Method | TL↓ | NE↓ | OSR↑ | SR↑ | SPL↑ |
| Plain | 11.20 | 4.52 | 60 | 55 | 52 |
| +Depth | 11.21 | 4.46 | 62 | 57 | 53 |
| +Disturbances | 11.93 | **4.12** | **66** | **60** | **55** |

We conducted an ablation study on the R2R-VLN task to evaluate the impact of various components in ActionVerse, including conversation-like data and different context lengths (Tables 3 and 4).

**Impact of Conversation-like Data.** In Table 3, we compare the performance of the model with and without conversation-like data. Without conversation-like data, the baseline model (**Plain**) processes single-step decisions using only RGB images and action candidates as input. Adding depth information (**+Depth**) improves the model's performance slightly.

When conversation-like data is introduced, the model maintains a history of observations and actions, which enables better multi-step decision-making. This improvement is reflected in enhanced performance metrics across the board (**Plain** with conversation-like data). Adding depth tokens (**+Depth**) further strengthens the model's performance. Additionally, incorporating data augmentation through disturbed routes (**+Disturbances**) enhances the model's ability to self-correct navigation errors, achieving the best results in NE, OSR, SR, and SPL metrics. However, this comes with a modest

increase in the average travel length (TL) as the agent backtracks to correct wrong movements, leading to longer paths.

Table 4: Ablation study on R2R-VLN task.

| Context Size | TL↓ | NE↓ | OSR↑ | SR↑ | SPL↑ |
|---|---|---|---|---|---|
| 512 | 11.23 | 4.60 | 64 | 54 | 53 |
| 1024 | 11.52 | 4.34 | 64 | 56 | 54 |
| 2048 (original) | 11.93 | **4.12** | **66** | **60** | **55** |

**Impact of Context Length.** The model's performance is also affected by the maximum context length, as it directly impacts how much history information can be retained during task execution. Our model operates with a context window of up to 2048 tokens, sufficient for most VLN tasks, where the typical conversation round involves about 10 turns. To evaluate the effect of different context lengths, we tested the model with shorter windows of 512 and 1024 tokens.

As shown in Table 4, performance declines with shorter context lengths, as vital information gets cut off, leading to reduced success rates (SR) and navigation accuracy (NE). The 2048-token context window retains the most historical information and achieves the best results across all metrics. This demonstrates the importance of maintaining sufficient context, particularly for tasks requiring rich historical reasoning, such as multi-turn navigation.

## 4.4 EFFICIENCY ANALYSIS

To demonstrate the efficiency of the proposed ActionVerse, we analyze the computational costs for both training and inference, highlighting the advantages in terms of implementation, deployment, and resource usage.

**Training Efficiency.** Our method leverages Vicuna (Chiang et al., 2023) as the base LLM and fine-tunes it using LoRA (Hu et al., 2022), a low-rank adaptation technique. This approach drastically reduces the computational requirements compared to training action models from scratch, which often demand large-scale resources. With our framework, the training can be completed on a setup of 4 Nvidia A100 GPUs in approximately one day, making it significantly more efficient than fine-tuning or training large, general-purpose multi-modal models. Furthermore, techniques like mixed precision training contribute to faster and more resource-effective training.

**Inference Efficiency.** For inference, we compare our method with NavGPT, an LLM-as-agent approach that uses ChatGPT APIs in a zero-shot manner. NavGPT requires elaborate system prompts of approximately 15,000 tokens per instruction, incurring a cost of around $0.75 per 10-round navigation request. Additionally, it depends on continuous network access to external API servers, which can be limiting in real-time applications. In contrast, ActionVerse eliminates the need for complex, manually designed prompts by aligning action tokens directly with the model's multi-modal space. This allows the system to be deployed locally on a single Nvidia A5000 GPU after quantization, significantly lowering the cost and making the model more accessible for real-time and on-device applications.

## 5 CONCLUSION

In this paper, we introduced ActionVerse, a framework that efficiently adapts chat-based MLLMs into action planners. Specifically, we designed a novel action encoding process alongside an alignment technique capable of simultaneously encoding both action types and action arguments into the foundational LLMs. By constructing a conversation-like instruction training set, we treated the action planning task as a dialogue between observers and planners, thus enabling an implicit self-reflection mechanism based on historical movements. We experimented with the proposed framework on two action-oriented tasks, demonstrating promising results with minimal instruction tuning. We anticipate that this work can serve as a baseline unified framework for a broader range of action-based tasks. Future work involves extending the proposed framework to more action-oriented tasks, such as smartphone app control, robotic system manipulation, and beyond.

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

# Appendix

## A  GENERATION OF CONVERSATION-LIKE INSTRUCTION DATA

In Section 3.1.2, we briefly introduced the conversation-like instruction-following dataset. Here, we provide more details.

Listing 1: Conversation-like Action Instructions

```
[
  {
    "system": {
      "content": "You are ActionVerse, a multi-modal planner. Based
          on the observer's input, first summarize the observations
          with keywords. Then, describe the reasoning process to
          achieve the target instruction: {instruction}. Finally,
          propose an action from the candidate pool."
    }
    "conversations": [
      {
        "role": "observer",
        "content": [
          "observation": "path/to/observation",
          "action_candidates": ["Move(viewpoint1)",
              "Move(viewpoint2)", "Stop()"]
        ]
      },
      {
        "role": "planner",
        "content": ["Observation: 1. Mudroom entrance with a window
            and light. 2. Spacious kitchen with stainless steel
            appliances. 3. Dining area with pendant lights. 4.
            pantry area with closed doors. \n Reasoning: To complete
            the task, I would seek out the entryway that likely
            leads to other rooms.\n Action: Move(viewpoint1)"]
      },
      ...
    ]
  }
]
```

Listing 1 provides an example of a generated conversation sample used for training the ActionVerse. Specifically, there are two sections: the first part is the *system* content, which specifies the general setting and the instruction. The *conversations* key stores the interaction between the observer and the planner. The observer's part contains the path to observations (in the navigation task, this refers to the RPG panorama of the room) and the possible actions that can be taken at the position. For example, there are two reachable points here, resulting in two `Move()` candidates and the `Stop()` action. The planner's part mainly includes three sections:

- Observation: Summarizes the main observations based on the observer's input.
- Reasoning: Describes the reasoning process based on the instruction and observations.
- Proposed Action: Suggests the appropriate action.

The main challenge in generating such conversations lies in the planner's part, particularly the detailed descriptions of the scene and the reasoning process. To obtain these descriptions, we rely on the ChatGPT-4v API. Specifically, we input a panorama image of the scene and ask GPT-4v to generate detailed descriptions of the scenario.

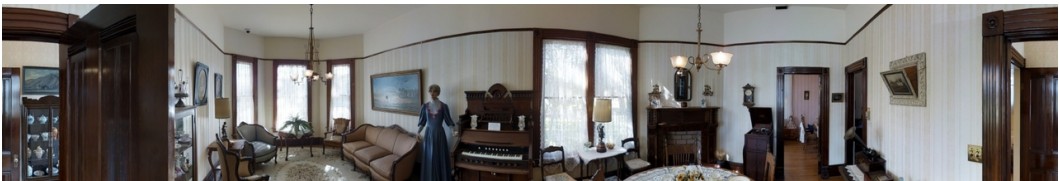

Figure 4: Panorama image of the scene.

The GPT-4v API demonstrates a strong capability to generate accurate descriptions of the scenes, which also contain directional information. Here is a sample output of Figure 4 from GPT-4v (we use green to annotate the correct concepts that appear in the image, blue for directional information, and red to highlight the incorrect output):

The image you've shared is a panoramic view of a vintage-style room, possibly from a historical home or museum, designed to replicate the interior from a past era, likely the late 19th or early 20th century. Here are the details:

- The room is spacious with high ceilings and features large, curtained windows that allow plenty of natural light to enter, contributing to a welcoming ambiance.
- The walls are adorned with striped wallpaper, and the woodwork, including the door frames, wainscoting, and crown molding, is dark-stained and highly detailed, indicative of the craftsmanship from the period.
- Traditional furniture pieces, such as a patterned sofa, armchairs, a round center table with a decorative vase, and a smaller side table with a vintage lamp, are arranged to create a comfortable seating area.
- A mannequin dressed in a period-appropriate dress stands near a pump organ, suggesting this room may be set up to reflect daily life from the era or for historical interpretation.
- To the left, there's a display cabinet filled with various antique dishes and glassware.
- The central area of the room features a large, ornate chandelier that matches the room's vintage aesthetic.
- In the middle of the room, there's a dining table set with a fine china tea set, as if prepared for a midday tea.
- The right side of the room houses a fireplace with a mantelpiece decorated with period items, including a small clock, a framed photograph, and other trinkets, further adding to the authenticity of the setting.
- A gramophone sits in the corner, and various framed pictures and paintings are hung throughout the room, completing the historical scene.

Overall, the room appears to be a meticulously curated space designed to give visitors a sense of stepping back in time.

We found that the GPT-4v model can accurately describe most of the concepts present in a scene. Based on these detailed descriptions, we used the GPT-4 API to summarize the observations into short sentences. Following instructions, we generated the reasoning process to achieve the goal. This method allowed us to create a complete conversation-like instruction training set for specific datasets and tasks.

# B    IMPLEMENTATION DETAILS

In this section, we provide a comprehensive explanation of our training process, hyperparameters, and model structures. The training of the proposed model follows a two-stage process:

- **Single-Step Instruction Tuning:** In this stage, we extracted approximately 28k single-step instructions from the R2R navigation training set. Each sample corresponds to a panorama photo of a specific location and a text instruction guiding the agent's intended direction. We generated observations and reasoning processes by providing the ground truth to ChatGPT, as detailed in Appendix A.1 of the original paper. The ground truth action served as the expected output.
- **Conversation-like Tuning:** The second stage fine-tunes the model with conversation-like data to enhance its chat capabilities and context understanding.

The entire training was conducted on a 4xA100 server over approximately one day. The detailed training parameters are presented in the following table:

| Configuration | Single-Step Instruction Tuning | Conversation-like Tuning |
|---|---|---|
| optimizer | AdamW | AdamW |
| epochs | 1 | 1 |
| effective batch size | 32 | 32 |
| learning rate | 2e-5 | 1e-5 |
| weight decay | 1e-4 | 0 |
| resolution | 448p | 448p |
| training samples (#instructions) | 28k | 31k |
| trainable param. | MLP, LLM(LoRA) | MLP, LLM(LoRA) |

Table 5: Training details.

Additionally, the LoRA configurations used for training are as follows:

| Configuration | Values |
|---|---|
| r | 32 |
| lora_alpha | 1 |
| lora_dropout | 0.1 |

Table 6: LoRA Configurations.

The network components are detailed in the following table:

| Vision Encoder | Text Encoder | Action Encoder | Projection Layer | LLM |
|---|---|---|---|---|
| DINOv2 | LLM Tokenizer | DINOv2 | MLP | Vicuna-7b |

Specifically:

- **Vision Encoder:** We use DINOv2 to encode the panorama scene image due to its capability with higher resolution images.
- **Text Encoder:** We utilize the pretrained tokenizer from the foundation large language model to encode the text instructions, following existing practices.

- **Action Encoder:** For action tokens, we added task-specific special tokens into the LLM vocabulary, such as `<move>` and `<stop>`. These tokens are encoded by the text encoder similar to other words. For the image region features inserted in special action tokens like `<move></move>`, we crop the regions from hierarchical feature pyramids of the last three layers from the image encoder.

- **Projection Layer:** Following LLaVA, we use a multi-layer perceptron to project the vision features into the foundation language model, initializing this projection layer using the LLaVA-1.5 pretrained model.

- **LLM** We employ Vicuna-7b as our foundation language model.

The proposed ActionVerse framework consists of four main stages:

- **Encoding:** Multi-modal information is encoded into tokens. This includes a DINOv2 vision encoder for image tokens, a pretrained LLM tokenizer for text tokens, and action tokens cropped from DINOv2's feature map based on action candidates.

- **LLM reasoning:** Non-textual tokens are aligned with the LLM space using MLP-based projection layers. All tokens, along with any historical information, are concatenated as input into the LLM.

- **Output detokenization:** The output tokens from the LLM are detokenized into natural language, containing observation, reasoning, and action, following training formats.

- **Execution:** The action token is extracted and sent to the navigation task simulator to move the agent or terminate.

---

**Algorithm 1** ActionVerse Process

---

1: **Input:** Status $S$, Text Instruction $T$, Action Candidates $A$, History $H$
2: **Output:** Status $S'$
3: **Initialization Stage:**
4: $I \leftarrow$ simulator.get_image$(S)$
5: **Encoding Stage:**
6: $I_{\text{token}}, A_{\text{token}} \leftarrow$ DINOv2$(I, A)$
7: $T_{\text{token}} \leftarrow$ tokenizer.tokenize$(T)$
8: **LLM Reasoning Stage:**
9: $proj_I \leftarrow$ MLP$(I_{\text{token}})$
10: $proj_A \leftarrow$ MLP$(A_{\text{token}})$
11: $input \leftarrow$ concat$(H, T_{\text{token}}, proj_I, proj_A)$
12: $output \leftarrow$ LLM$(input)$
13: **Output Detokenization Stage:**
14: $response \leftarrow$ tokenizer.detokenize$(output)$
15: **Execution Stage:**
16: $A' \leftarrow$ extract_action$(response)$
17: $S' \leftarrow$ simulator.make_effect$(A')$

---

