# OpenReview forum: "Action as a Modality: Turning Multi-Modal LLMs to General Action Planners"
_ICLR.cc/2025/Conference — ICLR 2025 Conference Withdrawn Submission_

### Official Review · Reviewer_BvPj · 2024-11-03

**Soundness:** 2
**Presentation:** 3
**Contribution:** 2
**Rating:** 5
**Confidence:** 3

**Summary:**

The paper introduces **ActionVerse**, a framework to adapt multi-modal LLMs for action planning by encoding action candidates into modality tokens. This approach reduces the need for complex prompts and aligns action spaces with LLMs' language space.

**Strengths:**

- The paper introduces ActionVerse, an innovative framework that adapts multi-modal LLMs to action planning by encoding action candidates as modality tokens. This approach removes the need for complex prompts, advancing task generalization in control applications.

- ActionVerse extends LLMs to real-time, action-oriented tasks without prompt dependence, broadening multi-modal LLM applications.

**Weaknesses:**

- The proposed action encoding method, while innovative, might encounter limitations when applied to tasks with large or continuous action spaces (e.g., in manipulation tasks, the action spaces are commonly continuous, like the 6D pose of end effectors). The action space in the experiments presented in this paper is limited to selecting from a set of camera IDs, which is significantly simpler than the action spaces encountered in typical robotics tasks. If the authors could validate their proposed method in a more complex action space, it would enhance the quality and impact of the paper.

- The writing quality could be improved, as there are expressions that lack precision and clarity. For instance, in lines 203-215, both sequence $S$ and observation $\mathcal{O}$ are ordered sets, so curly braces should not be used. Additionally, the angle brackets used in line 208 are not well-defined, which could lead to confusion. Clarifying these notations would improve readability and technical accuracy.

**Questions:**

- Can the authors elaborate on the generalizability of their action encoding approach to tasks with more complex or continuous action spaces? For example, how would ActionVerse handle real-world robotic manipulation tasks where the action space includes continuous controls, such as gripper angles or multi-joint motions?
- The paper mentions a conversation-like tuning method to support self-correction, yet it seems relatively basic compared to reinforcement learning or memory-based models that track long-term dependencies. Could the authors clarify any specific limitations they observed in the self-correction mechanism or discuss potential improvements they are considering?

---

### Official Review · Reviewer_4pR5 · 2024-11-04

**Soundness:** 4
**Presentation:** 1
**Contribution:** 3
**Rating:** 5
**Confidence:** 3

**Summary:**

This paper proposes ActionVerse, which aims to resolve the misalignment between the general pre-trained LLM and the action space of specific control tasks by encoding action candidates into a series of modality tokens, coupled with an efficient alignment technique to synchronize the action tokens with the LLM’s language space. The authors conduct experiments on robot navigation and website navigation, and show that their implementation of ActionVerse achieves performance comparable to SOTA methods.

**Strengths:**

1. This paper addresses an essential issue—integrating visual information and action space into LLMs, which is fundamental for using LLMs as decision-makers.
2. The approach shows improvements over previous LLM-agent methods across two important domains.

**Weaknesses:**

1. Some sections would benefit from reorganization, and certain techniques require further explanation. For example, projection layers are mentioned in lines 38, 74, and 142, but no explanation of what projection layers are until appendix. Additionally, it’s unclear why Algorithm 1 is located in the appendix. It would be helpful to have a clearly defined training and algorithm framework in the methods section to facilitate understanding.
2. The related work section describes prior studies in detail, but it lacks a clear positioning of this paper within that landscape and its relationship to those studies.
3. The techniques proposed require additional tokens or direct operations on embedding spaces, which might not be compatible with advanced LLMs, such as GPT-4. While these models offer strong reasoning capabilities, they often have constrained APIs.

**Questions:**

1. What tokens, and how many, are used beyond the vocabulary of the base LLM?
2. Is the argument embedding in line 315 a token? If so, how does it address the problem mentioned in line 59? If not, how is an embedding fed to the base LLM?
3. In lines 422-427, the authors claim that ActionVerse is more general compared to expert models. However, from lines 314-317, human-designed modules are still used for capturing visual representations. Could you elaborate on why ActionVerse is considered more general?
4. How are the planner and observer trained? Do you use supervised fine-tuning? What is the dataset size? How do you ensure that the dataset generated by GPT-4 will be beneficial for downstream navigation tasks?
5. Could you please compare the dataset size or training computational requirements to prior expert model methods? Is ActionVerse more efficient to train?
6. In line 535, the authors claim that the multi-turn conversation between the planner and observer represents "implicit self-reflection." Could you explain this concept further or provide an example?

---

### Official Review · Reviewer_wb8P · 2024-11-04

**Soundness:** 2
**Presentation:** 2
**Contribution:** 2
**Rating:** 3
**Confidence:** 3

**Summary:**

This paper presents a method for Vision-Langauge Navigation (VLN) tasks that attempts to improve upon the assumptions of prior work for how action encodings are created. In essence, the paper attempts to create a general-purpose action encoder, ActionVerse, which can encode actions as tokens for input to an LLM; the LLM can then serve as a planner over those actions. The system includes a prompting mechanism that facilitates an interaction between an observer and a planner to generate and execute the plan. The approach is validated on two benchmarking tasks: R2R-VLN and Web-VLN. The results are positive relative to other LLM baselines and mixed relative to expert baselines. Ablation studies are included.

**Strengths:**

- The description of the issues -with existing action encodings from Page 5-6 is helpful.

- The paper includes two reasonable tasks (R2R-VLN and Web-VLN) as well as a clear description of metrics.

- While the approach is outperformed by expert models, the approach does seem to outperform LLM-based baselines on the R2R-VLN benchmark. The approach outperforms all baselines on most metrics for the Web-VLN task.

- An ablation study with positive results is reported (Tables 3-4)

- Training and inference efficiency are discussed positively.

**Weaknesses:**

- In the sense of an MDP or POMDP, $s_i$ is typically a state and $a_j$ is an action. The use of the $\mathcal{A}$ for the set of actions but $s$ for individual actions is a confusing mismatch.

- Figure 2 does not sufficiently explain how the projection and action tokenization works. The figure could benefit from a legend and more annotations (e.g., what does (1), (2), and (3) refer to, where did they come from (as labels, for example, as an output of DINOv2 -- though the figure makes it look like an input), and where do they go?

- This example, "For example, the visual observations...robotic arms, etc." is insufficient to convince this reviewer that the framework is generalizable. The framework may work for simple move-to-area actions based upon DINOv2, but more work would need to be done to actually show generalizability to arbitrary embodied AI or robotic tasks. For example, how would this framework support these domains?

https://github.com/Healthcare-Robotics/assistive-gym
https://mujoco.org/

- Figure 3 does not clearly describe how the multi-round conversation plays out. Either this figure and/or the text needs to be expanded.

- The disturbance procedure for enhancing robustness needs to be further described.

- LLMs are not planners (see below). It would be helpful if the authors could better describe how they overcame the issues discussed in Kamghampati et al.

Kambhampati, S., Valmeekam, K., Guan, L., Verma, M., Stechly, K., Bhambri, S., Saldyt, L. and Murthy, A., 2024. LLMs can't plan, but can help planning in LLM-modulo frameworks. arXiv preprint arXiv:2402.01817.

- The approach seems relatively simple and has mixed results (not outperforming expert models on one of the two benchmarking tasks). The paper is also making strong claims about generalizability outside of simplistic VLN tasks. As such, the contribution seems limited and that the paper seems to be overclaiming.

- A running example for action encoding would have been more helpful, as that is the crux of the paper, and there is insufficient technical detail to precisely determine how this happens in general (e.g., a description of Figures 2 and 3 with sufficient technical detail to afford reproducibility).

**Questions:**

- Why did the paper not benchmark on the Web-VLN task the entire set of baselines from the R2R-VLN benchmarking task? Did the paper take those performance results from prior work or actually run an independent evaluation?

---

### Official Review · Reviewer_DgPF · 2024-11-04

**Soundness:** 2
**Presentation:** 3
**Contribution:** 3
**Rating:** 5
**Confidence:** 4

**Summary:**

This paper presents the ActionVerse architecture, which extends a multimodal large language model (LLM) to handle visual inputs and action outputs, enabling it to function as a policy within an environment. Unlike prior approaches that represent actions as tokenized outputs, ActionVerse encodes actions as inputs to the LLM and selects an appropriate action through a multi-round conversation between an observer and a planner. ActionVerse is evaluated against several baselines on R2R-VLN and Web-VLN and demonstrates superior performance compared to previous methods utilizing LLMs as policies.

**Strengths:**

1. ActionVerse presents a new alternative to prior works of LLMs for decision-making by encoding actions as inputs and then having the LLM choose between these inputs. This avoids the policy having to output any action representations that can be brittle and not robust relative to the LLM pretraining.

1. ActionVerse is empirically validated on two challenging tasks, showing strong performance against a range of baselines. It surpasses other methods that use LLMs as agents. On Web-VLN, it also outperforms approaches that rely on more specialized systems and non-LLM policies.

1. Ablations in Table 3 confirm the importance of the conversational-like data and the disturbance data augmentation. The impact of context length on the results is also displayed in Table 4.

**Weaknesses:**

In my assessment, the major novelty of this work is representing actions as inputs for an LLM rather than having the LLM output actions. Specifically, ActionVerse encodes actions via new tokens and encodes image inputs using features from a pretrained visual encoder like DINOv2. The LLM then selects which action to choose from all possible encoded actions. This contrasts with prior approaches that output actions directly from the LLM, for example, by tokenizing the action and then outputting the tokens corresponding to that action, as with RT-2. Encoding actions as input in the ActionVerse system poses an interesting alternative to such tokenized outputs. However, the paper does not evaluate if this different way for an LLM to output actions is empirically better than the direct tokenized output strategy of works like RT-2. To properly assess this key contribution, this work should have compared to the exact same system, using the same conversational data format, added disturbances, and all other aspects of ActionVerse, but then selected actions based on the LLM outputting tokens corresponding to the action as RT-2 does.

Instead, the paper conflates the need to maintain complex prompts for zero-shot LLMs with finetuning LLMs for actions. It is still possible to finetune the LLM to take actions by having the actions be output tokens as RT-2 does, and this does not require extensive system prompts or expensive calls to proprietary LLMs.

Overall, to properly assess the key contribution of ActionVerse in finetuning LLMs to encode actions as input, the paper must compare finetuning the LLM in the same setup to output actions instead.

Other weaknesses:

1. How can ActionVerse, which requires encoding all possible actions, extend to continuous action spaces or large discrete action spaces? In continuous action spaces, such a ranking scheme is infeasible. In large discrete action spaces, the input length would grow too large.

1. Given the performance of ActionVerse relative to the expert models, what is the value of using LLMs as agents in these problem specifications? In the VLN in Table 1, there is a large gap between ActonVerse and the expert models. The pretraining knowledge of LLMs should provide some benefit in terms of generalization.

**Questions:**

1. What benefits are LLMs as agents providing in Web-VLN and R2R-VLN, given they are outperformed or closely matched by non-LLM approaches? What additional assumptions do the best-performing non-LLM models require?

1. How does the usage of conversation-like data impact the context length during training and inference? While conversation-like data improves performance, does it come at a higher training and inference cost due to increased context lengths?

---

### Note · Authors · 2024-11-13

I have read and agree with the venue's withdrawal policy on behalf of myself and my co-authors.